# Egoism as a Problem for Robust Moral Realism

**Espen Ottosen** 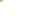

Fjellhaug International University College, Sinsenveien 15, 0572 Oslo, Norway; eottosen@nlm.no

**Abstract:** As a normative ethical theory, ethical egoism may seem compatible with the metaethical theory of moral realism. However, in this article, I will problematize such an assumption. The reason is that an important motivation for rejecting moral anti-realism by many moral realists—like Derek Parfit, Torbjörn Tännsjö, and Erik J. Wielenberg—is that such a view entails that not even cruel or horrendous acts are to be called wrong by any universal or objective standard. I suggest that this motivation also applies to the ethical theory of egoism, as it may imply that no one has any obligation to refrain from cruel or horrendous acts as long they are beneficial to the agent. On that basis, I will demonstrate that egoism is a problem for robust moral realists that also, to a large degree, is being overlooked.

**Keywords:** moral realism; metaethics; egoism; ethical egoism

## 1. Introduction

In this article, I will show that egoism is a problem for robust moral realism—a view that has become increasingly popular among moral philosophers in the last decade.[1] There are different kinds of moral realism, and moral realism may be described somewhat differently, but I take the main idea to be that "there is a moral reality that people are trying to represent when they issue judgment about what is right and wrong".[2] Some moral realists are naturalists, arguing that "a moral judgment is rendered true or false by a natural state of affairs".[3] However, in this article, I will discuss the view of those who claim that moral facts are not reducible to non-moral or natural facts.[4]

In recent years, philosophers like David Enoch and Erik J. Wielenberg have used the label "robust realism" to describe a metaethical position that is "an objectivist, non-error-theoretical, cognitivist, or factualist position, it states that some normative judgments are objectively non-vacuously true".[5] Although Parfit did not use this label and instead says that we "ought to accept some form of Non-Naturalist Cognitivism", his view fits this description well.[6] The moral realism of Parfit, Wielenberg, and Tännsjö implies that there exist moral facts that are objective and that moral statements should not be viewed as relative or subjective. In other words, moral normative reasons are not constituted independent of us.

Because Parfit has written extensively on the topic and "had for years been regarded as the best living moral philosopher" when he died in 2017 (McMahan 2021, p. 1; see also Chappell 2021, p. 1), his perspective will be my main focus. Although my argument should be relevant for most robust moral realists, I will primarily discuss—in addition to Parfit—the thinking of Eric J. Wielenberg and Torbjörn Tännsjö. Although some theists are also robust realists, my suggestion that ethical egoism represents a problem for robust moral realism is limited to explicitly non-religious realism, as theists have additional possibilities for rejecting egoism—for example, by arguing that God has commanded us to love each other.[7]

The version of robust moral realism that I will scrutinize in this article also subscribes to moral optimism in an epistemological sense. David Enoch argues that a moral realist at the same time can be a radical skeptic about moral knowledge.[8] Nevertheless, robust moral

realists generally think that "human beings have various moral obligations even if God does not exist".[9] They also hold "the view that our deepest moral beliefs are true, at least for the most part" (Killoren 2016, p. 226). Based on moral optimism, most robust moral realists criticize moral anti-realism for accepting or being passive in the face of morally repugnant acts (Tännsjö 2010, p. 57; Wielenberg 2017, p. 8; See also Shafer-Landau 2005, p. 5). This perspective is essential for Parfit, as he describes moral nihilism as "a bleak view" that "replaces goodness and badness with nothing". (Parfit 2011c, p. 190) As such, the moral realism of Parfit, and most versions of robust moral realism, "entails moral non-nihilism" (Killoren 2016, p. 226).

I will argue that ethical egoism—especially considering moral optimism—represents a possible defeater for robust moral realism. If moral anti-realism is to be rejected because such a view entails that not even cruel or horrendous acts are to be called wrong by any universal or objective standard, then it may be necessary to reject ethical egoism for the same reason. I further suggest that this problem of ethical egoism for robust moral realism has been overlooked because egoism—as a normative ethical theory—can be seen as compatible with the metaethical view of moral realism.

I will start out by discussing more precisely the problem of both psychological and ethical egoism. Then, I will explain why robust moral realists should provide compelling arguments for rejecting these forms of egoism. The last part of my article discusses how robust moral realists respond to ethical egoism. I will show that moral obligation toward other people is mainly presupposed and that some of the arguments used against egoism do not work.

## 2. The Problem of Psychological Egoism

It is not controversial to state that people, in general, are, at least to some extent, motivated by egoistic assessments. We are all preoccupied with our own situation, trying to avoid trouble, misery, and suffering, and looking for satisfaction and happiness. That is why Ivar Russay Labukt contends that the answer "because it is in your own interest" is "the only kind of justification of morality that is available" (Labukt 2015, p. 81).

One common way to reject psychological egoism is by arguing that some people in some circumstances seem to be motivated by genuine altruism. Brad Hooker argues that "psychological egoism overstates the truth that very many human beings are largely motivated by self-interest" and goes on to say that "the high percentage of people who regularly make uncoerced intentional sacrifices for the sake of others must be shocking to anyone who started off thinking that either psychological egoism or something close to it is true" (Hooker 2013, p. 713). See also Shaver (2021).

I do not think that this rejection of psychological egoism is sufficient. Although some persons may be genuinely altruistic and make intentional sacrifices, I also think that psychological egoism gets much explanatory power from an overwhelming number of cases where horrible things are done to others—by hurting, exploiting, and sometimes killing them—and the most obvious explanation is that people try taking care of their own interest. According to Roger Crips, "it is self-evident that if some action promotes my well-being, I have a reason to do it proportional in strength to the degree of promotion" (Crisp 2006, pp. 71–97, 126).

My point here is not to argue for the truthfulness of psychological egoism on a general basis. I will limit myself to arguing that the fact that many of us in many circumstances are motivated by egoistic inclination represents a challenge to anyone subscribing to moral realism, hence also robust moral realism. That is because the fact creates the need to explain how and why a widespread instinct or inclination may be, at least in some circumstances or situations, contrary to moral facts.[10] In other words, we should be able to demonstrate that there really exist binding moral reasons like, for example, "one should not treat other people in (...) an inhuman way (Österberg 1988, p. 168).

As a theist, referring to a God who has given every person an obligation to express love toward others is a viable option. An atheistic version of robust moral realism needs another

kind of explanation for why rejecting some of an agent's egoistic impulses is warranted. Bloomfield points out that it is a problem for those rejecting egoism in that it is not at all obvious "why people should accept the authority of morality over self-interest" (Bloomfield 2014, p. 12), which, for me, is a reason to link our obligation toward the well-being of other people together with a theistic moral perspective.

Further, the plausibility of psychological egoism makes some common responses to ethical egoism a bit off the mark. Although the argument, made famous by Joseph Butler, that "I must desire things other than my own welfare in order to get welfare" may often be true, it is not obvious that I should avoid horrendous acts if I am confident that such acts benefit my welfare.[11]

I think there are relevant arguments against universalizing egoism. It is not consistent for me as an egoist to think that everyone has the right to pursue his own interests without any limits, as those interests may conflict with my own well-being.[12] However, I do not think that such an argument nullifies the strength—and hence the problem—of psychological egoism, as it makes a lot of sense for a psychological egoist to be agent-biased or have an agent-relative perspective.[13] However, I will go on to discuss how egoism as an ethical theory also represents a problem.

### 3. The Problem of Rational or Ethical Egoism

Many introductions to moral theory use a couple of pages—sometimes a chapter or two—on ethical or rational egoism as a normative ethical theory, at times presented as a form of consequentialism.[14] However, not many contemporary philosophers will embrace egoism as a sustainable normative ethical theory. In an introduction to moral theory, Russ Shafer-Landau writes: "It may seem there is nothing 'ethical' at all about such theory" (Shafer-Landau 2015, p. 107). Brad Hooker explains that egoism "is often taken to be one of the main enemies of morality" (Hooker 2013, p. 710).

Basically, ethical egoism may be defined as the view that "one should promote one's own good" or "we have no duties to anyone but ourselves".[15] For my discussion, it is not important to distinguish between rational egoism, viewed as a perspective on rational behavior, and ethical egoism as a specific moral theory, as both theories reject, at least initially, any other obligations than an agent's well-being.[16] So, when referring to ethical egoism in continuation, I am including rational egoism as well.[17]

The main issue, as I see it, concerns the implication of rational or ethical egoism. Is it a matter of fact that egoism means that it is ethical, as Russ Shafer-Landau writes, "secretly killing political opponent, stealing from the weak, or humiliating their employees"?[18]

In this context, I would point out that it is not problematic per se that an agent looks out for his self-interest, well-being, or prudence.[19] I don't presuppose or argue that following egoistic desires is always wrong, especially when others are not negatively affected.[20] The discussion about balancing prudence and morality is comprehensive, and I am inclined to follow Roe Fremstedal that, even though there is a substantial overlap, there is also a "limited conflict between morality and prudence" (Fremstedal 2018, p. 12). It may also be possible that, as Labukt suggests, "egoistic considerations in most cases support a commitment to morality that is fairly deep and at least as extensive as the one displayed by most actual people" (Labukt 2015, p. 81).

Many different arguments have been used to explain why ethical egoism does not entail the acceptance of cruel or horrendous actions so far as it benefits the agent. For example, it does not follow that ethical egoism implies subjectivity on self-interested reasons. Hence, one could argue that a horrible or ruthless act will not really pay off in the long run,[21] or highlight that a person could be wrong about what is in their interest (Bloomfield 2008a, p. 9).

Terrence Irwin describes Aristotle as "a psychological egoist" who also "relies on rational egoism, thinking that each rational agent has overriding reason to do what best promotes his own happiness" (Irwin 2007, pp. 125, 132). However, Aristotle also "demands

for the virtuous person to elect the virtuous action for its own sake" (Irwin 2007, p. 188; see also p. 206). So, I am skeptical to view him as an ethical egoist and prefer the term Eudaimonist.[22] Similarly, Paul Bloomfield defends a kind of ethical egoism that rejects what he calls immortality since "happiness and self-respect require respecting others appropriately".[23]

It is beyond the scope of this article to discuss all the arguments, perspectives, and strategies that have been put forward to explain why an ethical egoist should or could not be cynical or amoral. Although I accept that some forms of ethical egoism suggest that an agent should be moral or virtuous, I still think that there is an inherent problem facing ethical egoism simply because this moral perspective links moral obligation together with an agent's self-interest. Let me try to substantiate that claim briefly by relating to a well-known illustrative story by Peter Singer:

> Imagine you come across a small child who has fallen into a pond and is in danger of drowning. You know that you can easily and safely rescue him, but you are wearing an expensive pair of shoes that will be ruined if you do. We all think it would be seriously wrong to walk on past the pond, leaving the child to drown, because you don't want to have to buy a new pair of shoes—in fact, most people think that would be monstrous. You can't compare a child's life with a pair of shoes! [24]

Confronted by this story, one might say that it is reasonable not to save the child, as it takes time and resources, may involve some risk taking, and could ruin expensive clothes and shoes. Those reasons are egoistic, of course, all referring to self-interest. However, there are also egoistic reasons to save the child—like being considered a nice person, not being haunted by a bad conscience, receiving gratefulness from the child and parents, etc.

I think that most people would save the child, perhaps for different reasons. And some ethical or rational egoists will argue that it is morally obligatory to rescue this child. Any other view is referred to as a form of ethical egoism that is "cynic", "immoral", or "amoral".[25] However, those who reject immoral or cynical egoism should explain why those who do not want to do so—perhaps quite a small group—are wrong. If we do not have such a reason, the obvious consequence is accepting that some people, perhaps psychopaths or sociopaths, think that a strange child is not their business.

In our world, it seems like some people do not care about people other than family and friends at best. They exploit, attack, and deceive others because of selfish motives. In such situations, selfishness clearly trumps empathy. Anyone who has become very rich—for example, through human trafficking—seems to have concluded that the personal gain outweighs the suffering of strangers.

Some would say that "honesty is a virtue because it is necessary for man to live successfully on earth" (Locke and Woiceshyn 1995, p. 410), but I do not find such a perspective to be plausible in a general sense. Even though an agent may find it sensible to be honest in most situations, the story of the child in danger of drowning makes it difficult, within the framework of ethical egoism, to explain why an agent is *obligated* to help or rescue other people, even strangers. It is not sufficient to point out that very few people are prepared to neglect the well-being of others. It is necessary to argue that they violate an ethical reality of some sort.[26]

Furthermore, when confronted by situations where it seems grossly immoral to be passive, it seems absurd to argue on the terms of ethical egoism "that helping benefits me".[27] Hence, I think that rational and ethical egoism is confronted by a fundamental problem when asserting as a starting point that there are *no other duties* than egoistic duties.

Gensler explains that ethical egoism implies doing "whatever maximizes their own self-interest, regardless of how this affects others" (Gensler 1998, p. 144). Even if some proponents of egoism—perhaps quite many—think that it is problematic or unwise to do cruel or horrendous acts, I do not think that there exists a sustainable barrier against such behavior for those accepting that no duties other than egoistic duties exist. At least it seems

like a possibility that some persons—perhaps sociopaths or psychopaths—reasonably think that, in some circumstances, they will be far better off by acting cruelly or ruthlessly.[28]

This is further supported by plausible examples of people who disregard the situation of other people in pursuit of their best interests. One such story is told by James Rachel about a doctor charging a black vulnerable woman a lot of money even though he had not really helped her.[29] After discussing the story, Österberg concludes that, although "it is in a person's interest to treat his fellow beings decently", it is also very plausible that "for certain people in certain circumstances quite horrible actions do in fact pay off" (Österberg 1988, p. 81). Thus, it is difficult to see that, within the framework of ethical egoism—where the only obligation is caring for oneself—it can be ruled out that it is never ethically acceptable or rational to do cruel, horrendous, or ruthless actions.

To conclude, I find it plausible that ethical egoism implies the moral acceptability of cruel or horrendous acts so far as an agent thinks that this will be beneficial for his or her well-being. I think Brad Hooker is correct when pointing out that the "standing possibility of conflict between morality and self-interest couldn't be a truism if morality simply required that people maximize their own good".[30] Even though some ethical egoists reject the acceptability of cruel or horrendous acts, and they may have good arguments, I think I have substantiated that ethical egoism represents a problem for robust moral realists who motivate their rejection of moral anti-realism by referring to the importance of condemning cruel or horrendous acts. My point is not to defend or reject egoism as such, but to point out why moral robust realists should address the problem of egoism. On this, I will now elaborate further.

## 4. Why Robust Moral Realism Needs to Reject Egoism

At first sight, it may seem like ethical egoism as a normative moral theory may be combined with moral realism as a metaethical position.[31] In the context of defending moral realism, Torbjörn Tännsjö argues that utilitarianism (or "something close to utilitarianism") probably is the correct moral theory but simultaneously does not rule out that ethical egoism may be the correct theory (Tännsjö 2010, pp. 8, 39).

However, I think such an assumption is incorrect when considering why many robust moral realists reject moral anti-realism. As a relevant illustration, let us look at how Tännsjö writes about torture. When commenting on the possibility of gaining moral knowledge, he uses the torture of an innocent child as an example:

> It should be noted that when I claim that it is obvious that it is wrong to torture an innocent child 'for no reason', this additional 'no-reason' clause is crucial to my argument. According to some ethical theories, held by clever people who have thought hard about these things, it is indeed all right to torture an innocent child—if the consequences are good enough to compensate for the suffering felt by the innocent child. I have myself argued that it might be right to torture an innocent child, if this means that a great many people, who are all of them already very happy, will undergo a sub-noticeable increase in their respective hedonic situation.[32]

Hence, for Tännsjö, the torture of an innocent child could be justified if such an act leads to positive consequences for a lot of people. At the same time, he writes that, if he meets a person who argues that it may be acceptable to torture a child *for no reason*, he will "set their opinions to one side; their moral opinions are just bizarre, and we try to find an explanation to do with some disorder or distortion within the dissenting person".[33]

Now, I suggest that Tännsjö's argument also applies to ethical egoism in general. Although they do not accept the torture of an innocent child for no reason, the theory may imply—as I have argued—that a sufficient reason is that an agent will gain some (subjective) satisfaction or well-being. This is because it is plausible that some persons (probably rather few) have a condition that means they would enjoy torturing a child. I find it obvious that

such a reason for torture should be viewed just as bizarre—and repugnant—as torture without a reason.

Derek Parfit argues that it is not rational to "save ourselves from one minute of discomfort rather than saving a million people from death or agony" and goes on to say that such a "horrendous act would not be rational".[34] However, it is difficult to rule out that such an act may be acceptable, and hence rational, for an egoist. Therefore, accepting ethical egoism as a plausible moral theory implies that it may be ethical to perform horrendous actions. This is a troublesome conclusion for any moral realist who is also a moral optimist. Hence, demonstrating that there are some objective moral facts does not make much of a difference if this principle may be the truth of ethical egoism.

Perhaps Parfit would respond by saying that he rejects such an ethical perspective, as he is not an ethical egoist. But, since he discusses repeatedly when it may be morally acceptable to follow egoistic inclination and when it is not, he demonstrates that ethical egoism is to be taken seriously. It may be said that ethical egoism conflicts with common-sense morality,[35] but, in Ancient Greece, "Altruism was not an ethical ideal, and Egoism therefore not a controversial position which has to be defended" (Österberg 1988, p. 15). So, rejecting ethical egoism because it is self-evidently wrong is not a viable option. Further, as the robust moral realists mentioned in this article try to refute moral anti-realism in their works, it seems highly relevant to scrutinize their responses to ethical egoism.

One of those who discussed the force of ethical egoism rather thoroughly was Henry Sidgwick, the highly influential British utilitarianist from the nineteenth century. "It is reasonable for a man to act in the manner most conducive to his own happiness", he wrote (Sidgwick 1996, p. 119). At the same time, he looked for reasons to reject—or at least limit—ethical egoism, as he also found self-evident "that the good of any one individual is of no more importance than the good of any other".[36]

Sidgwick then found two competing ethical principles that he attempted to unify somehow. Time after time in *The Methods of Ethics*, he looks at this challenge from different angles, and, at one time, he called it "the profoundest problem of Ethics".[37] To refuse ethical egoism outright seemed impossible, as "when we sit down in cool hour, we can neither justify to ourselves this or any other pursuit till we are convinced that it will be for our happiness, or at least not contrary to it" (Sidgwick 1996, p. 120).

It is widely recognized that Sidgwick did not succeed in unifying ethical egoism with the utilitarian idea that everyone has an obligation to seek the happiness of all.[38] As Derek Parfit is deeply influenced by Sidgwick, it is not surprising that he discusses the same tension.[39] He rejects the view of Sidgwick that it is equally rational to be an egoist as to prioritize the happiness of others by stating that no one has a sufficient reason to avoid mild discomfort if the alternative is to rescue a lot of people.[40]

As is evident, it is important for Parfit to reject ethical egoism, as it may accept horrendous acts. If egoism is an acceptable theory, a person can neglect famine, war, rape, or genocide if such acts or tragedies do not affect his well-being somehow. Such an implication is upsetting to Parfit. He writes that other people's suffering should matter to every one of us.[41] For decades, it has been central to his moral thinking that "there are certain people to whom we have special obligations" (Parfit 1979, p. 556). In addition, we have some obligations to strangers.[42] This is also the perspective of Wielenberg, who argues that anyone has "normative reasons" for helping someone in pain (Wielenberg 2017, p. 6).

Stating that there is a moral obligation to help everyone in pain, even when it is not beneficial to the helper, implies a rejection of egoism. Such a perspective thus demonstrates how ethical egoism is a problem for robust moral realists. It is not obvious that an egoist will view horrendous acts as objectively or universally immoral, and the robust moral realist needs to explain why he holds the opposite position. This is also important, as everyone seems motivated, at least often, by his or her own well-being. In such a situation, it is necessary to explain why it may be immoral to prioritize our own interests.

Now, I will support my perspective by demonstrating that some arguments against moral egoism in fact appeal to egoistic inclination and, hence, support the theory rather than weaken it. I will also argue that the possibility that most of us are hardwired, because of evolution, to care about other humans poses an additional problem that robust moral realists need to address when facing egoism.

### 4.1. Arguments That Reveal the Strength of Egoism

Erik J. Wielenberg writes explicitly that it is immoral for an agent to think purely in egoistic terms. Instead, we should be concerned with others' situations for moral reasons. "The fact that rape harms its victim in serious ways is a compelling reason for me not to commit rape, regardless of whether refraining from rape somehow benefits me."[43] When trying to back up this view, Wielenberg writes:

> Being moral often facilitates the development of meaningful connections with other people, whereas moral transgressions often tend to isolate us from each other. Extreme immorality may produce substantial material benefits; however, the cost of such immorality—even when undetected—is often social isolation, or at least decreased connectedness with others. There are some obvious ways in which this tends to be true: most people don't like to be around jerks, much less form close relationships with them. [44]

Basically, I agree with Wielenberg. At the same time, I think his reasoning could make egoism more plausible as an ethical theory. Arguing that it is unwise to be an egoist, in the long run, is, in fact, an argument that simply grants egoism as true. The claim that it pays off to care about others—at least when taking an overall perspective—is an appeal to human selfishness. What is missing, then, is explaining why one should care about another person's situation, even if it is not in their personal interest to do so.

This perspective is also acknowledged, at least implicitly, when Parfit writes: "We can have strong reasons to care about the well-being of certain other persons, such as our close relatives and other people that we love" (Parfit 2011a, p. 40). These strong reasons show why egoism makes so much sense: A reason to care for people close to us is that it benefits us in different ways. To boil it down: when they are happy, we are happy.

As I see it, a moral argument then is—in an important sense—one or more reasons why we should *not* act by egoism. Therefore, I also think a fundamental part of moral reasoning consists of identifying reasons that override self-interested reasons. This is also the implicit understanding of Wielenberg when he says, referring to Kant, that "taking duty to be an overriding consideration is a distinctive feature of a morally good person, whereas taking one's self-interest to be the most important consideration is a distinctive feature of a morally bad person".[45] Also, Parfit thinks that egoism should not be regarded "as a moral view, but as an external rival to morality".[46]

At the same time, Parfit writes that "we are morally permitted to give some kind of strong priority to our own well-being" (Parfit 2011a, p. 149). I agree that it is acceptable to take an agent's well-being into account, but such a conclusion makes it relevant to question why it is not morally acceptable to always follow egoistic desires. Hence, it is necessary to identify specific moral reasons that, at least in some circumstances, make it wrong to prioritize our own well-being. As we will soon see, Parfit discusses different thought experiments about how much priority should be given to egoistic consideration, but he does not explain why egoism sometimes is immoral. However, I will now move on to explaining why evolution makes it even more important to identify moral reasons for refuting egoism as an acceptable moral theory.

### 4.2. If Both Moral and Immoral Instincts Are Hardwired

When discussing rational egoism, Shafer-Landau writes that anyone has a reason to help or warn a person who may be run over when crossing a street, a young woman dragged into a dark alley by "a gang of boys", or a dehydrated hiker (Shafer-Landau

2005, p. 196). Such a reason could be explained by the fact that our human empathy is an evolutionary trait. However, stating that an overwhelming majority of people think (or feel) that we should help a stranger is not a compelling argument that we are more obligated to help others than ourselves. As moral realists also admit that evolution may have given us some problematic moral instincts (like racism or tribalism),[47] we need an argument for stating that it is morally necessary, and not just a common instinct, to care about other persons even if it is not beneficial to the agent.

The discussion about moral realism considering evolution is comprehensive. Both Parfit and Wielenberg discuss the influential article by Sharon Street arguing that moral realists face a "Darwian Dilemma" (Street 2006; Parfit 2011b, pp. 525–42, 2011c, pp. 264–90; Wielenberg 2005, pp. 152–56). According to Street, evolutionary forces make it doubtful that we can detect moral truths "as tendencies to make certain kinds of evaluative judgments rather than others contributed to our ancestors' reproductive success not because they constituted perceptions of independent evaluative truths, but rather because they forged adaptive links between our ancestors' circumstances and their responses to those circumstances, getting them to act, feel, and believe in ways that turned out to be reproductively advantageous" (Street 2006, p. 127).

It is not easy to get a clear grip on Parfit's response to Street. He seems to simultaneously admit and deny that human morality is a result of evolution.[48] However, his main response is that there is a "Darwian answer" to the dilemma, and it consists in the fact that evolution has given humans the capacity to reason. "The ability of early humans to form such true beliefs had evolutionary advantages, by helping them to survive and reproduce".[49] And the ability to reason may also be used to evaluate, and reject, moral instincts or ideas that are unreasonable.

However, this argument does not present any reason to reject egoism as such. I do not deny that the ability to reason is helpful and important, but to get any *genuine moral discussion* off the ground, we need to explain why it is not morally acceptable to reason only to protect my interests. Two critics of Parfit's argument, Peter Singer and Katarzyna de Lazari-Radek, argue that they have a better answer than Parfit because "utilitarians are at an advantage over those who hold moral views that are based on our commonly accepted moral rules or intuitions" (Singer and de Lazari-Radek 2017, p. 286). Nevertheless, they are confronted by the same challenge because they do not justify the utilitarian principle of maximizing utility for everyone. To refute egoism, we need to know why such an obligation is objective.

Wielenberg argues that rationality (developed by evolutionary mechanisms) has ensured the development of the idea of human equality. Unfortunately, he presents no convincing arguments for such a claim. In one place, he writes that "the relevant cognitive abilities ensure a correlation between moral rights and beliefs about moral rights because they *entail* the presence of moral rights and create beliefs about such rights" (Wielenberg 2017, p. 155). I find this sentence to be either meaningless or circular.

Furthermore, Wielenberg grounds the idea of all people's rights in the fact that evolution has constructed human beings so that we unconsciously classify other people as "like" ourselves.[50] But such optimism seems unsustainable. It is easy to imagine that some—perhaps also in our time, but at least in the past—reject that certain others are identical to them; perhaps because they have a different skin color, a different culture, or belong to a different caste. In such a situation, it seems absurd to suggest that evolution created a belief in equality.

In addition, a possible counterargument to Wielenberg could be to state that evolution may have produced empathy, also toward strangers, but that such an instinct should be rejected, as it is not rational to care for others—as long as it is not beneficial to the agent. In the face of such logic, a solid counterargument is needed, and I cannot see that Wielenberg provides that. Moreover, the idea of human equality need not be an objective, moral principle, even if evolution should have given us this conviction. It cannot be ruled out that evolution may have equipped us with the wrong idea.

## 5. The Responses of Moral Realism to the Challenge of Egoism

So far, I have argued that egoism—both as a descriptive explanation of human behavior and as an ethical or rational theory—represents a problem for robust moral realism. This fact is also explicitly acknowledged by Derek Parfit. Now, I will proceed by mainly discussing how Parfit and Wielenberg relate to this challenge. I will show that some of their arguments to reject egoism are inadequate. Further, I will show that Parfit to a large degree presupposes the existence of moral obligation toward other people instead of offering arguments for such a position and thereby rejecting ethical egoism. My aim in this section is not to present any argument from silence, but rather to substantiate the problem of egoism by showing that this reality is not adequately addressed by Parfit and Wielenberg.

### 5.1. Inadequate Responses to Egoism as a Challenge

With regard to Sidgwick (1996, p. 167), Parfit discusses the idea that there never is—in fact—any real conflict between being egoistic and being impartial (and therefore caring about *everyone* affected by an act). However, he rejects such a solution by ascertaining that at least sometimes caring for other people comes at a personal cost (Parfit 2011a, pp. 137, 382).

In *Reasons and Persons*, Parfit explicitly criticizes rational egoism. He proposes the following counterexample to such a theory:

> *My Heroic Death.* I choose to die in a way that I know will be painful but will save the lives of several other people. I am doing what, knowing the facts and thinking clearly, I most want to do. (...) I also know that I am doing what will be worse for me. If I did not sacrifice my life, to save these other people, I would not be haunted by remorse. The rest of my life would be well worth living. (Parfit 1984, p. 132)

However, this argument does not explain why it is *morally wrong* to be a rational egoist but rather argues that some people may have a sufficient reason not to act as rational egoists.[51] In that sense, Parfit argues like a subjectivist, only showing that someone has a subjective reason for preferring to save other persons. But the main moral question remains to be answered: is anyone obligated to save another person's life even at the expense of his own life?

In a pragmatic sense, Parfit discusses the importance of caring for other persons' interests in many different thought experiments in *On What Matters*. When doing so, he does not identify any general principle that explains when caring for others, including strangers, should outweigh the individual's well-being. In fact, Parfit seems to jeopardize morality as such by arguing that, in some rare circumstances, we could have "sufficient reasons" to act immorally (Singer and de Lazari-Radek 2017, p. 284). de Lazari-Radek and Singer conclude that Parfit "undermines morality to a significant degree" by not finding a solution to the dualism between egoism and universal benevolence (Parfit 2011a, p. 135).

As mentioned, Parfit thinks it is horrendous not to be willing to "save ourselves from one minute of discomfort rather than saving a million people from death or agony".[52] I agree. But why is it not, according to Parfit, horrendous—or immoral—to save one's own finger instead of another life, or to decide to save one's own life even when sacrificing it could save two thousand lives?[53] Parfit writes that it would be rational, but immoral, for a man to steal medicine from a stranger to save his own life and two children when he knows that a stranger needs the medicine to save her own life and her four children".[54] However, he does not explain why morality in this situation should override the rationality of self-interest. Parfit's different examples do not help us in identifying any objective reason why egoism should sometimes be rejected in favor of others' well-being or why moral obligation overrides egoistic reasons.[55]

"We all have reasons to regret anyone's suffering, and to prevent or relieve this person's suffering if we can", writes Parfit (Parfit 2011a, p. 138). Nevertheless, it does not look like he has identified any arguments that, confronted with the rationality of egoism, make those reasons into objective, moral obligations. Where Sidgwick found two rival

ethical theories—rational egoism and utilitarianism—and did not see the possibility of letting one of them win through, it is less clear what position Parfit considers as plausible (Crisp 2021, p. 151), but when his ethics becomes practical, egoism gets a lot of priority.

Wielenberg also rejects that a person's self-interest will always coincide with morality.[56] However, he simply states that being moral is a duty: "To the question "why be moral?" a perfectly acceptable answer is "because it is moral". This might seem odd until one notices that to the question "why do what is in one's interest?" a perfectly acceptable answer is "because it is in one's interest'".[57]

I do not think this argument can escape the charge of begging the question. Further, an appeal to egoism does not need to stand alone, as we can provide answers like "Because it will make you happy", "Because it avoids pain", or something like that. As Kurt Baier highlights when explaining why rational egoism is "highly plausible", everyone is expected to present a reason for not doing something that seems to conflict with their own interest (Baier 1991, p. 201). Therefore, we need a positive and compelling account of the existence of objective obligations toward other persons.

Wielenberg, perhaps more explicitly than Parfit, presents common-sense morality as a fundamental reality that does not need any justification. He writes that a human being has ethical obligations, unlike animals, because they can "evaluate, suffer, experience happiness, explain the difference between right and wrong, choose between right and wrong, and set goals for itself, has specific rights, including the right to life, liberty and the pursuit of happiness" (Wielenberg 2017, p. 56). As I see it, this is a puzzling statement since Wielenberg refers to empirical truths that are not controversial (e.g., that humans can reason, suffer, and experience happiness) alongside presupposing both the existence of rights and moral truths. In some sense, this means that Wielenberg chose to neglect all arguments put forward by non-cognitivists and anti-realists of different strands.[58]

### 5.2. Presupposing Moral Obligation Instead of Arguing

Through hundreds of pages, Derek Parfit insists in *On What Matters*—and to some extent also in *Reasons and Persons*—that moral anti-realism is untenable. His starting point, and I think it is fair to say that it is also his main argument,[59] is that the reasons for our actions should not be viewed as purely subjective, based on personal preferences that cannot be evaluated or criticized. As an important argument, Parfit asks us to imagine a specific man:

> Consider first an imagined man who has an attitude that we can call *Future Tuesday Indifference*. This man cares about his own future pleasures or pain, except when they will come on any future Tuesday. This strange attitude does not depend on ignorance or false beliefs. Pain on Tuesdays, this man knows, would be just as painful, and just as much *his* pain, and Tuesdays are just like other days of the week. Even so, given the choice, this man would now prefer agony on any future Tuesday to slight pain on any other future day. [60]

This example leads Parfit to conclude that "this man's preferences are strongly contrary to reason and irrational".[61] On that ground, he argues that reasons may be evaluated by some objective and rational standards.[62]

For Parfit, thinking in terms of reason also applies to morality. He explains that something matters only if we or others have some reason to care about it. "It would have great importance if morality did not in this sense matter because we had no reason to care whether our acts were right or wrong. To explain and defend morality's importance, we can claim and try to show that we do have such reasons".[63]

However, the matter of fact is that Parfit does not present any specific arguments that morality—as objective duties and obligations—really exists. He tries primarily to show that moral reasons cannot be natural and argues against subjectivism and nihilism. I have not identified any positive arguments in *On What Matters* that count in favor of robust moral

realism, although he seems to acknowledge (as in the citation just presented) that such arguments are much in need.[64]

After rejecting that reasons can be purely subjective and presenting some initial moral reflections, Parfit states in *On What Matters* that he will not say much "about these *meta-ethical* questions" besides in Part Six (Parfit 2011a, p. 174). At the same time, he also writes that "our moral theories are primitive and have grave defects",[65] perhaps indicating why he is not able to provide reasons to believe in objective duties or obligations. When returning to metaethical topics in Part Six, he mainly keeps arguing against other positions and refrains from positively supporting his position through arguments.

The consequence is that he—a lot like his master Sidgwick—presupposes that objective moral obligations toward others exist (perhaps thinking that they in some senses are self-evident). It seems paradoxical that Parfit, who often argued in *Reasons and Persons* that common sense should be challenged, in *On What Matters* explicitly refers with support to "the overlapping sets of beliefs that most people accept, which Sidgwick calls *common-sense morality*" (Parfit 2011a, p. 149) and generally presupposes that widely shared moral opinions are correct.

Interpreting Parfit generously, we may say that his arguments in favor of normativity in general, primarily by rejecting subjectivity about reasons, count in favor of moral realism. But even so, a significant problem is that those arguments do not rule out the truthfulness of ethical egoism. Even conceding that it is irrational for a person to be indifferent to any suffering on a given Tuesday, this argument primarily supports egoism because it refers to the good or bad of the agent and no one else. Precisely therefore we need an *additional* reason for claiming that no one should be indifferent to other people's pain, even when that pain does not directly affect them.

According to robust moral realism, moral truths, duties, and obligations really exist. But they are not natural truths available for natural science to determine. Hence, J. L. Mackie famously remarked back in 1977: "If there were objective values, then they would be entities (. . .) of a very strange sort, utterly different from anything else in the universe (Mackie 1990, p. 38). According to David Killoren, robust moral realism should be viewed as a religion—even an excellent region—because "they endorse beliefs in non-natural moral facts" instead of demonstrating the existence of such facts (Killoren 2016, p. 230).

The response to Mackie from Parfit is among his most controversial. He argues that moral truths "have no positive ontological implications" and exist in a "non-ontological sense".[66] Chappell argues that Parfit therefore "seeks to defang such metaphysical qualms by denying that objective values [. . .] would have to exist 'in the universe' at all" (Chappell 2021, p. 8). It is worth noting that such claims explain moral facts by saying what they are not: neither non-natural nor ontological. Responding to these qualifications, Alan Gibbard writes: "I don't myself know what non-natural, non-ontological properties are or what a non-causal 'response' is, though Parfit says some things on this, and perhaps intelligible explanations can be given. As I read Parfit, he doesn't claim to know either".[67] David Copp suggests that Parfit's theory should be called a "minimalist theory of moral truths" simply because it explains very little.

Parfit seeks to point out similarities between mathematical or other abstract realities to back up his robust moral realism (Parfit 2011b, pp. 475–87). But it is not at all clear if such a perspective explains the essence of moral truths, as mathematical truths have the advantage of seeming more self-evident, and often can be demonstrated logically, than moral truths. Those truths are not confronted by the plausibility of egoism in different versions.

## 6. Conclusions

It is plausible that some normative statements are objective in the sense that not complying with them is irrational. The empirical fact that smoking tobacco statistically will bring about sickness and a reduced life span is also a reason not to smoke. As I see it, only those arguing that the pleasure of smoking outweighs the health damage may reasonably continue to smoke. In this sense, Derek Parfit—and other robust moral realists—have a

compelling reason to reject or weaken a subjective perspective on reasons and normativity. There is something deeply counterintuitive in rejecting the notion that science, surveys, and statistics could help identify something universal about how to avoid suffering and promote well-being.

Nevertheless, there is a startling difference between saying that a particular action is harmful to the agent and saying that it is morally wrong because it is harmful to another person. The first statement is egoistic. For those committed to securing their own well-being—as most of us are—it is reasonable to do whatever makes life better. In many situations, that implies doing something good for other people. Such action may bring about gratefulness, some nice feelings, and often a reciprocal response. Nevertheless, this is no other thing than some form of egoism.

What robust moral realism needs—to be able to reject horrendous acts as immoral—is a convincing argument that we have a moral obligation toward other persons that is objective, universal, and does not depend on the possibility of any personal satisfaction. I have tried to show that the argumentation put forward by the robust moral realists Derek Parfit, Erik J. Wielenberg, and Torbjörn Tännsjö do not provide such arguments. Instead, they mainly presuppose that there exists such an obligation. For me to see, this reveals a vulnerability of robust moral realism that would be solved by a theistic version.

**Funding:** This research received no external funding.

**Conflicts of Interest:** The author declares no conflict of interest.

## Notes

1   A recent survey shows that more philosophers in 2020 than in 2010 call themselves moral realists. The number has gone from 56% to 62%; see Bourget and Chalmers (2014, p. 476); and Bourget and Chalmers (2023). It is also worth noting that Peter Singer, the undoubtedly most well-known moral philosopher today, wrote in 2017 that he once was an ethical subjectivist but had become a moral realist. "There are, as Derek Parfit has argued in his major work On What Matters […] objective ethical truths that we can discover through careful reasoning and reflection", wrote Singer (2017, p. xii).

2   (Shafer-Landau 2005, p. 13). See also Miller (2013, pp. 1–5), who identifies moral realism with strong cognitivism; Sayre-McCord (2021).

3   (Miller 2013, pp. 3–4). Miller also names Nicholas Sturgeon, Richard Boyd, David Brink, Richard Brandt, and Peter Railton as naturalist realists. Among other normative naturalists is Copp (2017).

4   For some thinkers, this is important, and, in On What Matters, Parfit (2011b, pp. 263–377; 2011c, pp. 65–98) uses a couple of hundred pages to argue why moral facts must be non-natural. Parfit also writes in *On What Matters (Volume 2)*, p. 267, that naturalism is "close to Nihilism". This is viewed as completely unreasonable by Copp (2017, p. 28).

5   (Enoch 2011, p. 4). Wielenberg refers to Enoch and supports what he calls "robust normative realism" but also presents his view as "Godless Normative Realism" (Wielenberg 2017, p. 14). Also, Shafer-Landau (2005, p. 8), defends "the view that moral principles and facts are objective in a strong sense" and that they "are not scientific ones".

6   (Parfit 2011b, p. 264). In volume three of "On What Matters", Parfit (2011c, p. 56) uses the term "Non-Realist Cognitivism" on his metaethical position, but there is no doubt that he upholds his moral realism. (Killoren 2016, p. 224; Hooker 2021, p. 227). It is also worth noting that Parfit declared: "Moral truths are not true only for certain people" (Parfit 2011c, p. 420). I will come back to the semantic of Parfit later.

7   This is also the view Parfit expresses on the possible relationship between God and ethics (Parfit 2011a, pp. 165–66), although he also rejects that normative facts can be grounded in claims about God (Parfit 2011b, p. 444). A theistic moral realist is Terrence Cuneo (2007).

8   (Enoch 2011, pp. 4–5). Therefore, I concede that my argument in this article does not fully apply to the view on Enoch.

9   (Wielenberg 2005, p. 68). Shafer-Landau (2005, p. 231) thinks that not many moral realists are moral skeptics in an epistemological sense, as this would be "a position of last resort, however, anyone taking such a stand bear the burden of explaining his confidence that there are such truths, while maintaining that these truths are unknowable".

10  As I see it, this is also the reason why rational egoism is "the 'default view' that any rival normative theory must defeat" (Shaver 1999, p. 1). Also, Sidgwick, as we will see, thinks that looking out for one's own happiness is rational for every human.

11  (Shaver 2021); see also Hooker (2013, p. 711). For more about the Ethical Egoism of Butler, see Österberg (1988, pp. 24–27).

12  (Gensler 1998, p. 144). A very similar argument goes back to G. E. Moore. "Egoism". See also Nagel (1978, p. 86) for the same argument. See also Österberg (1988, pp. 85–87).

13  See Brink (1997, pp. 107–10), for a discussion on this.

14    See Gensler (1998, pp. 143–45). There are also some philosophers who explicitly present consequentialism as something different from egoism (Shafer-Landau 2015, p. 119).

15    (Österberg 1988, p. 1; Tännsjö 2008, p. 40; Bloomfield 2014, pp. 13–14). According to Shafer-Landau (2015, p. 106), ethical egoism means that there is only one ultimate moral duty—"to improve your own well-being as best as you can". (Gensler 1998); Shaver (2021) writes that ethical egoism means "do what maximizes your self-interest".

16    What rational egoism does, compared to ethical egoism, is avoid "controversies about how relativistic a moral theory can plausibly be and about what the deep explanation of moral wrongness is" (Hooker 2013, p. 716). Shaver (1999, pp. 2–4) presents a similar distinction, although he also follows Sidgwick in thinking that rational egoism is viewed as "a theory of reasons that competes with traditional moral theories". See also Crisp (2021, p. 151).

17    See Österberg for a comprehensive presentation of many different forms of Ethical Egoism (Österberg 1988, pp. 35–48). See also Bloomfield for a presentation of different forms of ethical and rational egoism (Bloomfield 2008a, pp. 3–9).

18    Shafer-Landau (2015, p. 107). Also, Hooker (2013, p. 714) refers to "a case where someone would benefit from killing his political or professional or romantic rival" and firmly concludes that those actions "would of course be wrong". See also Crisp (2006, p. 135).

19    Many important moral theories, however, reject such a perspective, arguing that "agents must not see their own interests, or the interests of their families, communities, etc., as having any special standing whatsoever in the decision procedure". This includes Kantian deontology and consequentialism. (Bloomfield 2008a, p. 3).

20    According to Österberg, it is "to act prudently, not egoistically" to promote "one's interest when they do not conflict with those of other people" (Österberg 1988, p. 3).

21    Therefore, Österberg is discussing thoroughly different views on "time-neutral egoism" (Österberg 1988, pp. 57–68).

22    (Fremstedal 2018, p. 13). Tom Angier (2018, p. 257) argues that "scholars like Julia Annas, Jennifer Whiting, Terrence Irwin and Howard Curzer have pressed the case for a non-egoistic reading of Aristotle". See also Annas (2008, p. 220), who explain that "aiming at flourishing by living virtuously does not make me egoistic in any sense".

23    (Bloomfield 2014, p. 9). Also, rule ethical egoism may be used to limit the problem of grossly immorality done by ethical egoists (Hooker 2013, p. 714). Roger Crisp (2006, p. 135) explains that an agent should consider the well-being of others partly because of our "evolutionary background and an emotional make-up which cannot be ignored in ethics".

24    (Singer 2009). Singer also finds that "I would have sufficient reasons, for example, to suffer an injury in order to save the life of a stranger" (Singer and de Lazari-Radek 2017, p. 281).

25    (Bloomfield 2008b, pp. 254–55; Locke and Woiceshyn 1995, pp. 405–6; Paley 2022, pp. 22–23) presents an interesting example of immoral egoism, based on moral anti-realism. He argues that "morals are for suckers", as striving to be moral is being held back. To get rid of moral ideas will make a person happier and more successful.

26    I chose not to discuss moral particularism, although such a perspective may weaken my claim; see, for example, Dancy (2004).

27    (Shaver 2021). Labukt underlines that ethical egoism explains that "I should not set fire to innocent persons because doing so is ultimately bad for me" and finds that this is not a satisfactory justification, (Labukt 2015, p. 93; see also Hooker 2013, p. 715).

28    This point is acknowledged by Bloomfield (2008b, p. 261) when writing about immoral egoists or "pleonetics" who "do not value what they do not know or even want, namely true love or real friendship". It may be, as Bloomfield argues, that those persons are self-deceiving, but it is difficult to see that such a conclusion follows from the principle of ethical or rational egoism.

29    Referred to by Österberg (1988, p. 80).

30    (Hooker 2013, p. 715); Österberg labels those believing that "there is not opposition between (their respective versions of) Ethical Egoism and (conventional) morality for "soft egoists" (Österberg 1988, p. 3). As will be evident, I am skeptical of this position, as are all the moral robust realists I am discussing.

31    It should also be pointed out that some find ethical egoism plausible because they adhere metaethically to anti-realism (Österberg 1988, p. 79).

32    Tännsjö, p. 51.

33    Tännsjö, p. 57.

34    Parfit, p. 135.

35    This is presented as one counterargument by Shaver (2021).

36    Sidgwick, p. 382.

37    Sidgwick, p. 386, n.4; Bloomfield (2014, p. 3) thinks that the solution is to view those principles as "interdependent", but it is outside the scope of this article to discuss his response to Sidgwick.

38    (Baier 1991, pp. 43–44); Parfit describes Sidgwick's view in this way: "Suppose next that one possible act would be impartially best, but that some other act would be best for ourselves. Impartial and self-interested reasons would here conflict. In such cases, we could ask what we had most reason to do all things considered. But this question, Sidgwick claims, would never have a helpful answer" (Parfit 2011a, p. 133). Sidgwick (1996, p. 411) also admits that he does not have a solution to the "the profoundest problem".

39    Parfit writes that he has two masters, Sidgwick and Kant, but that he thinks that "Sidgwick's book contains the largest number of true and important claims." (Parfit 2011a, p. xxxiii).

40    Parfit, p. 365.

41    Parfit, p. 138.

42    Parfit, p. 556.

43    Wielenberg, p. 57.

44    Wielenberg, pp. 57–58.

45    (Wielenberg 2005, p. 78). The reference to Kant is interesting, as Kant argued that moral obligation in its deepest sense needed grounding in the existence of God, which is what Wielenberg tries to reject. Later, he rejects this argument of Kant mainly by presupposing the existence of moral obligations that "the universe is only as just as we make it, and consequently there is a much greater urgency to pursue justice here on earth". pp. 80–89.

46    (Parfit 2011a, p. 166). Parfit also writes that it "is unclear, for example, whether our reasons to promote the well-being of others should all be called moral reasons" p. 167. See also Österberg, who writes that the egoistic perspective that considering the interests of other people is not an obligation means to reject "the received conception of morality" (Österberg 1988, p. 1).

47    (Singer and de Lazari-Radek 2017, pp. 290–91). Singer and de Lazari-Radek, and Singer, suggests that the intuitive negative reaction toward incest, even among adult siblings, may be a result of evolutionary forces that today should be rejected (Parfit 2011b, p. 536).

48    He writes that our cognitive abilities "were partly produced by evolutionary forces" (Parfit 2011b, p. 520). But he also rejects that "normative beliefs were mostly produced by evolutionary forces". Parfit, p. 534.

49    (Parfit 2011b, p. 494). This argument is also supported by Singer and de Lazari-Radek (2017, pp. 286–87).

50    Wielenberg, pp. 145–46.

51    Therefore, Chappell underlines that the argument of Parfit primary amounts to "a competing view that is either more objective or else more subjective" (Chappell 2021, pp. 2–3).

52    (Parfit 2011a, p. 414). Jakobsen discusses thoroughly how Parfit thinks about the possibility of viewing moral reasons as overriding other reasons and concludes "that Parfit thinks we generally have most reason to do our duty—that is, moral rationalism generally holds—but that we might not always have most reason to do our duty" (Jakobsen 2020, p. 52).

53    Instead, Parfit writes: "I might perhaps have such reasons whether my injury would be as little as losing one finger (...) I would have sufficient reasons to save either my own life or the lives of several strangers. And I might have such reasons whether the number of these strangers would be two or two thousand". He also writes that "I could rationally save one of my fingers rather than saving some stranger's life". Parfit, p. 38; p. 40.

54    This example is from a personal correspondence referred to in Singer and de Lazari-Radek (2017, p. 283).

55    Parfit acknowledges that if "we often had decisive reasons to act wrongly, that would undermine morality". But he does not rule out that we may have such reasons occasionally (Parfit 2011a, p. 147); Jakobsen points out that Parfit never says that moral reasons override other reasons (Jakobsen 2020, p. 91).

56    (Wielenberg 2005, pp. 70–77). However, it is important for him to show that this line of thinking was important to Hume and Aristotle and is worth considering.

57    Wielenberg, p. 79.

58    Wielenberg also claims that "some ethical facts" fall into the category of being "brute facts", and, therefore, it is misguided to ask for their foundation or "where do they come from". Wielenberg, p. 38.

59    Which is why Scanlon (2014, p. 2) calls the position of Parfit "reason fundamentalism".

60    (Parfit 2011a, p. 56). An identical example is provided in Parfit (1984, pp. 123–24).

61    (Parfit 2011a, p. 56). As Chappell (2021, pp. 5–6) argues, our ends are, according to Parfit, "open to rational evaluation".

62    In fact, Parfit is more or less baffled by the fact that "Subjectivism about Reasons are now very widely accepted" and, thus, uses many pages to try to find out why such an erroneous theory has such an appeal (Parfit 2011a, pp. 65–70).

63    Parfit, 148. He also writes that "[f]or morality to matter, we must have reasons to care about morality, and to avoid acting wrongly" Parfit, p. 147.

64    This is also the conclusion of Jakobsen (2020, p. 125).

65    Parfit, p. 174. He also says that "Non-Religious-Ethics is at a very early state" (Parfit 2011c, p. xiii).

66    (Parfit 2011b, pp. 479, 481). See also Jakobsen (2020, pp. 107–20), for a thorough survey of the semantics and arguments of this topic by Parfit.

67    (Gibbard 2017, p. 61). See also Jakobsen (2020, p. 120) for a presentation on how this view of Parfit has baffled fellow philosophers.

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
