# Peer review of "Egoism as a Problem for Robust Moral Realism"

_religions, doi:10.3390/rel14101315_

Round 1
Reviewer 1 Report
11) Is the aim of this paper mainly to argue that moral realism contradicts rational/ethical egoism or, rather, that moral realism is a stronger, more defensible view than rational egoism is? The latter claim requires a comparative account that shows that egoism is weaker than moral realism. The former, on the other hand, must show that egoism cannot be reconciled with moral realism (as suggested by, for example, Aristotelian higher-order egoism in which moral virtue is prudentially necessary for eudaimonia, cf. Irwin 2011: vol i on Aristotle and rational egoism). But either claim requires far more precision to be defensible. Specific forms of egoism contrast with moral realism, yes – but which is it exactly? Instead of targeting rational egoism more generally, it is better to target freeriding, non-universalized variants that are amoralistic or immoral. For it is such amoral variants that are central to the literature on egoism and amoralism or practical moral skepticism (see Bloomfield 2016: 13-14).
22) Distinguish clearly between practical moral skepticism regarding practical reasons, and epistemological moral skepticism concerning reasons for belief or knowledge (or theoretical skepticism concerning statements and facts). The term “moral skepticism” could cover both, although Superson (2009: 3) and Schaber (2015: 34) use it in the former sense, something that corresponds to Sinnott-Armstrong’s “practical moral skepticism” (2019a, b).
33) I’d suggest that you focus on practical moral skepticism, sometimes dubbed amoralism or even immoralism, and how it relates to rational/ethical egoism. In the literature, practical moral skepticism is normally, but not always, based on rational egoism. Other forms of amoralism or practical moral skepticism are indeed possible conceptually but seem far less significant normatively.
44) In any case, the account of ethical/rational egoism must be far clearer. Specifically, distinguish between derivative and nonderivative reasons for action and between simple and sophisticated higher-order (split-order) views that are egoistic on one level but not another (cf. Brink 1997a,b; Irwin 2011: vol I). Also, rational egoism concerns not just duties or requirements but also what is rationally permissible. Specifically, it disallows net self-sacrifice that is often associated with moral demandingness.
55) Consider the metaethical or meta-normative implications of amoralism (practical moral skepticism). Specifically, distinguish between objective (agent-neutral) conceptions of self-interest and subjective (agent-relative) conceptions of self-interest. Explain why amoralism is virtually universally based on the latter, not the former, and how this contrasts with moral realism. Despite your claims on page 3, rational egoism need not be about “subjective satisfaction,” since there are many different accounts of egoism, prudence, and personal happiness.
66) Here is one suggestion: objective (agent-neutral) conceptions of self-interest generate problems for rational egoism, at least in common amoral, freeriding forms (cf. Hills 2003). Specifically, asserting “each man’s happiness” as “the sole good,” involves a clear contradiction (Moore 1903: 99; Nagel 1978: 86). For “each man’s happiness” entails a plurality of goods that contradict “the sole good”. To avoid this problem, one could deny individuality and collapse personal interest with the common good, something that clearly undermines freeriding, non-universalized amoralism. Or one could argue that everyone ought to only serve the self-interest of one specific individual, something that is an indefensible position that arbitrarily favors one specific individual over all others (van Ingen 1994: 39-48; Hills 2003; Sterba 2014: 36-8). Finally, interpreting self-interest in objective, agent-neutral terms means that self-interest is considered impersonal. There is no special reason then for me to aim at my self-interest – there is only reasons for all of us to aim at an objective good (cf. Nagel 1978: 123-4). Alternatively, there is reason for all of us to aim at several objective goods. But there is no reason to favor one individual over another, unless we introduce agent-relative, subjective conceptions of self-interest. For objective (agent-neutral) conceptions of self-interest do not support amoralism as subjective (agent-neutral) conceptions of self-interest and egoism do (by claiming that one should care only for one’s personal good, interests, or happiness). Specifically, common non-universalized, freeriding, and egoistic variants of amoralism require agent-relativity that favors oneself. At least in part, this seems to be the reason that why amoralism is commonly combined with subjectivism regarding self-interest, not objective conceptions of it.
77) Distinguish between moral reasons, prudential/self-interested reasons, and reasons for action (or living) that concern practical reason, all-things-considered. Is rational egoism wrong about all of these three or only the two latter ones? Isn’t it possible with hybrid accounts, such as that of Sidgwick, that see moral reasons as altruistic and prudential ones as egoistic, something that allows normative conflict that undermines reasons about what to do, all-things-considered? Do such conflicts involve a clash between agent-neutral morality and agent-relative self-interest?
88) Note that morality need not override competing reasons – it can silence them instead, either normatively or motivationally (or moral reasons could be defeated by prudential ones). The important thing for your purposes is mainly normative priority or rational authority.
99) To criticize rational egoism, consider the two following options. First, to argue for the “paradox of egoism” according to which altruism serves self-interest better than egoism does (something that makes rational egoism self-defeating unless it is weakened significantly). Second, discuss in more detail counter-examples to rational egoism that typically concern justified net self-sacrifice. Hooker explains:
Rational egoism entails that, in a case where one is deciding between what is most beneficial to oneself and what is beneficial to others, one could not rationally decide to pass up what is most beneficial to oneself. [….] Rational egoism is mistaken as long as there is a rational permission to sacrifice one’s own good for the sake of some other things. And that such a rational permission obtains seems overwhelmingly plausible. Hence, after careful attention to the examples where a very small sacrifice for the agent is needed to bring about a huge benefit for someone else, rational egoism seems extremely counter-intuitive. (2013: 716-17; cf. Carson 2012: 144-5)
110) Note that rational egoism represents an extreme form of partiality that differs from weaker forms of partiality towards near and dear. Hooker explains:
PARTIALITY involves assigning more importance to the welfare or will of some individuals or groups than to the welfare or will of others. Egoism is an extreme form of partiality, in that it gives overriding importance to just one individual’s welfare [or will]. There are different kinds of impartiality, but the kind of impartiality that juxtaposes with egoism and partiality is impartiality towards the welfare or will of each. (2013: 710)
111) Moral rationalism and anti-rationalism are clearly relevant here. But it a little too much for this paper perhaps. Often, anti-rationalism appeals to a weak form of egoism, such as veto egoism, that places limits on acceptable self-sacrifice though.
112) Clarify the relationship between values and reasons, normatively.
Bloomfield, Paul (2016[2014]) The Virtues of Happiness: A Theory of the Good Life, Oxford: Oxford University Press (Oxford Moral Theory).
Brink, David (1997a) “Rational Egoism and the Separateness of Persons,” in Jonathan Dancy (ed.) Reading Parfit, Oxford: Blackwell, 96-134.
(1997b) “Self-Love and Altruism,” Social Philosophy and Policy 14: 122-57.
Carson, Thomas (2012) Lying and Deception: Theory and Practice, Oxford University Press.
Hills, Allison (2003) “The Significance of the Dualism of Practical Reason,” Utilitas 15: 315-29.
Hooker, Brad (2013) “Egoism, Partiality, and Impartiality,” in Roger Crisp (ed.) The Oxford Handbook of the History of Ethics, Oxford: Oxford University Press, 710-28.
Irwin, Terence (2011) The Development of Ethics: A Historical and Critical Study, vols. I-iii, Oxford: Oxford University Press.
Nagel, Thomas (1978) The Possibility of Altruism, Princeton: Princeton University Press.
Moore, G.E. (1903) Principia Ethica, Cambridge: Cambridge University Press.
Schaber, Peter (2015) “Why Be Moral: A Meaningful Question?,” in Beatrix Himmelmann and Robert Louden (eds.) Why Be Moral?, Berlin: de Gruyter, 31-41.
Sinnott-Armstrong, Walter (2019a) “Moral Skepticism,” Stanford Encyclopedia of Philosophy, ed. by Edward Zalta,
https://plato.stanford.edu/entries/skepticism-moral/ (2021/07/28)
(2019b) “Practical Moral Skepticism,” Stanford Encyclopedia of Philosophy, ed. by Edward Zalta, https://plato.stanford.edu/entries/skepticism-moral/supplement.html (2021/07/28).
Sterba, James (2014) From Rationality to Equality, Oxford: Oxford University Press.
Superson, Anita (2009) The Moral Skeptic, Oxford: Oxford University Press (Studies in Feminist Philosophy).
Author Response
Thank you for a very relevant and comprehensive response. I have made significant revisions, especially in the first part of my article, as the comments revealed the need for more precision. Many of the references in the response have also been consulted. Let me then comment more specifically point-by-point:
11) My main point is to argue that rational/ethical egoism (and to some degree also psychological egoism) is a problem for robust moral realism and has to be addressed. I have tried to substantiate this point by discussing different forms of rational and ethical egoism and then argue that the main perspective of egoism is a problematic one for those thinking our moral responsibility towards other people is essential (as most moral robust realists think).
22) I have almost everywhere changed the term "moral sceptisism" to "moral anti-realism".
33) I have tried to explain why there is an immoral or amoral potential within mainstream ethical egoism or rational egoism although I fully acknowledge (and refers to) that there are ethical or rational egoists with other persecitives.
44) I agree and have made many changes to accommodate the need for a more precise account of ethical/rational egoism.
55) I have tried to explain that my main focus is the non-universalized version of egoism -- which I at the same think is somewhat stronger and more important than often recognized.
66) In the revised article I am discussing explicitly the argument of Moore/Nagel. And I have also made other changes to the problem of non-universalized amoralism (which I argue could be seen as a part of egoism more generally).
77) I realize that I did not distinguish properly between prudence and amoral egoism, and I have made changes to be more precise.
88) Interesting point, and indirectly addressed, I hope, in my revised version.
99) In different parts of the article I discuss if -- and to what extent - it is helping a person's self-interest to be altruistic. I still think such an argument is an appeal to egoism -- and try to explain why I have my reservations!
110) Thank you for an interesting quote by Hooker which I now have read and used in different places in my revised article.
Reviewer 2 Report
This paper makes a number of interesting points and raises a number of cases regarding the tension between acting in your own self-interest vs acting in the interests of others. However, there is a serious definitional problem that overrides these values in this article.
I would like to suggest that you have incorrectly characterized (or defined) the terms "Ethical Egoism or Rational Egoism". What you are describing is better referred to as "Cynical Egoism". You also seem to switch between “egoism” and “ethical egoism” as though they are the same thing. They are not.
You say "the interesting feature of both rational and ethical egoism is the view that an agent has no other obligation than his/her well-being." and "Now, I suggest that Tännsjö’s argument also applies to ethical egoists. Although they don’t accept the torture of an innocent child for no reason, the theory implies that it is a sufficient reason that the agent will get some (subjective) satisfaction. It is plausible that some persons (probably rather few) have a condition that means they will enjoy torturing a child. I find it obvious that such a reason for torture should be viewed just as bizarre – 97 and repugnant – as torture without a reason." And “As is evident, Parfit sees the importance of rejecting ethical egoism as it may accept horrendous acts.” And “An egoist has no reason for viewing horrendous acts as objectively and universally immoral…”
If this were true - if ethical egoism or rational egoism really meant this - then I would agree with your argument that this is indeed a lousy thing. However, absolutely nobody who would claim to be an ethical egoist or rational egoist would agree with you here that it is OK to torture a child just for their own pleasure. This view mischaracterizes what it means to be an ethical egoist or rational egoist.
This is a Straw Man Fallacy - you set up a misleading or vile definition of rational egoism and then demonstrate that it is a lousy idea that is not in harmony with Moral Realism (or any other ethical theory).
I see that you use several references that appear to give support for your definition of how potentially rotten an ethical egoist could be. There are a large number of contradictory, and far better, references that you should consider if you want to have a fair and balanced perspective that does not try to rely upon the Straw Man Fallacy.
Any fair discussion of egoism should recognize that the entire reason why scholars have added the term "ethical" or "rational" to the term "egoist" is precisely because people like to claim that egoist can do whatever they want for their own subjective pleasure. What you are referring to is “cynical egoism” (e.g. see Locke, E. A., & Woiceshyn, J. (1995) not “ethical or rational egoism”.
There is a long line of rational egoist scholars who provide contradictory definitions to the one proposed in this article. This would include Adam Smith who stated that pursuing one’s self-interest is bound by morality and justice and that an individual should not advance their own interests to the detriment of other people. Smith advocated justice through the rule of law, human rights, property rights, economic liberty, and the enforcement of contracts that facilitate a level playing field (Sen, 1999; Werhane, 2000). Smith emphasized the importance of reciprocity to free markets (James & Rassekh, 2000), benevolence (Locke, 1988) and mutual gain (Johnson, 1995).
I know she is not very popular in academia, but Ayn Rand, perhaps the leading advocate for rational egoism, stated: “Man—every man—is an end in himself, not a means to the ends of others; he must live for his own sake, neither sacrificing himself to others nor sacrificing others to himself; he must work for his rational self-interest, with the achievement of his own happiness as the highest moral purpose of his life.” She very clearly stated that one cannot simply do whatever one wishes and that it is profoundly in one’s long-term self-interest to be honest and not harm others (Certainly she would never say that torturing a child for one’s own pleasure could be moral!).
Locke and Becker (1998, p. 170) state that “... rational egoism holds that principles and values are good to the extent that they are in an individual's long-term, rational self-interest - that is, to the extent that they promote a person's survival and well-being over the course of a lifetime.”
I could potentially suggest many references, but here is the one that seems to best describe what you are referring to in your article. Locke, E. A., & Woiceshyn, J. (1995). Why businessmen should be honest: The argument from rational egoism. Journal of Organizational Behavior, 16(5), 405-414.
I urge you to have a fair, balanced and more thorough description of rational or ethical egoism and contrast these with cynical egoism. That way you would avoid the Straw Man Fallacy and provide a more nuanced and accurate description of this term. I also think it would make the cases you describe later in the article more interesting and relevant.
Author Response
I take that the main point for this reviewer is to point out that I am not presenting well enough the different forms of ethical and rational egoism. As explained by my point-to-point response to reviewer 1, I have made substantial changes in order to be more precise.
Reviewer 3 Report
A well-written piece. Congratulations.
Author Response
Thank's
Round 2
Reviewer 1 Report
The paper has been somewhat improved, but a few issues remain and should be dealt with.
1. Rational Egoism comes in many forms, but the author tends to focus only on subjectivistic, non-universalized and amoral forms of it. But it crucial that things would look very different if one either moralizes egoism, as Irwin does by arguing that prudence requires virtue, or if one is a realist about self-interest. The first option rules out normative conflict regarding morality and prudence. The second shows that the distinction between realism and anti-realism/subjectivism cuts across the distinction between egoism and morality. Egoists need not be subjectivists regarding self-interest, since it is perfectly possible to be mistaken about one’s self-interest (indeed it is possible to go even further, particularly if one moralizes self-interest as Irwin and others do). Ethicists on the other hand need not be realists – they could be subjectivists or antirealists or constructivists. The paper needs to be clearer on this issue.
2. The concept of objectivity is too unclear. Objectivity in the sense of being mistaken differs from objectivity in the sense of impartiality as well as objectivity that is agent-neutral. Which of these three senses are central to ethics and which are central to self-interest?
3. Moral anti-rationalism is a big issue that this paper touches on without any proper discussion. For anti-rationalism, an act can be immoral yet rational, all-things-considered. Oftentimes, anti-rationalism appeals to some sort of weak egoism such as veto egoism that puts limits on acceptable self-sacrifice. But anti-rationalism is often not an egoistic theory normatively (even though broadly Aristotelean variants are common).
4. The paper confuses ethics with universal principles, ruling out moral particularism without any argument.
If these issues are dealt with and clarified, it would be much easier to see exactly how egoism poses a problem to moral realism.
Some proofreading seems necessary
Author Response
Thanks for an important and relevant response. I have tried to revise my article according to the suggestions.
(a) I have underlined more thoroughly that there are different forms of ethical egoism which will not accept horrendous or horrible actions as long as an agent finds that they increase his/her well-being. I have also tried to explain more precicely why I find that ethical egoism faces an inherent problem as the theories starting point is that every ethical obligation is connected to the well-being of the agent. My view may be a bit controversiel, but I think have been able to present relevant arguments.
(2) I refer to objectivity as a metaethical perspective opposed to subjetive (or relativistic) perspectives, and have underlined that explicitly.
(3) I think it will take need many paragraphs to discuss anti-rationalism as such, but when discussing the overridingness of morality, especially related to the view of Parfit, I am implicitly showing that the robust moral realists I am discussing, are not subscribing to anti-rationalism. And that, I hope, is sufficient.
(4) I am no longer talking about universal moral principles, and do also refer in a footnote to moral particularism.
Reviewer 2 Report
This quick revision is much appreciated and the author has bolstered his or her arguments with a number of new citations and added some level of nuance. But the author has not really amended the document to address my key concern.
I must admit that I am perplexed and somewhat disturbed that the author has refused to accept the suggestion to reference a few key scholars that make a few important points that should be included in this discussion. Locke, in particular, is an obvious and important reference who has almost 170,000 citations and an i10 index of almost 300. (I note that none of these suggested references are my own.) Adam Smith is also a relevant foundational figure in the topic of rational egoism who is conspicuous by his absence.
The author thus continues to make statements that do not acknowledge the contributions of the important references I pointed out. For example, the author states:
“Harry J. Geisler argues that it is not possible to hold on to ethical egoism consistently because such a theory implies that “we’d have to desire that X harm us greatly (even paralyze us for life) if this would maximize X’s self-interest”.26 [this makes no sense to me. Perhaps this is a typo?]
and
“However, one important aspect of psychological egoism consists in overlooking other people’s interests. That means it makes a lot of sense for an egoist to think that his only desire regarding the behavior of X is that X does not harm him – and therefore he will do what is necessary to avoid such acts.”
And
“The main issue, as I see it, concerns the implication of rational or ethical egoism. Is it a matter of fact that egoism means that it is ethical, as Russ Shafer-Landau writes, “secretly killing political opponent, stealing from the weak, or humiliating their employees”?
Most rational egoists or ethical egoists (and all of the references I suggested) would strongly disagree with these statements. They would say that you must consider the full range of context (not ONLY their own interest) and that harming others for your own selfish gain is NOT moral. They would certainly NOT say that one should overlook other people’s interests.
It is simply not “a matter of fact” that egoism means the things that the author claims. There are important scholars who directly refute the author’s assertion. I would urge the author to appropriately acknowledge these contributions to the literature.
I am concerned that the author is cherry-picking only the authors that reinforce their own personal opinions rather than openly accepting that certain important and well-known scholars may disagree with their opinions. Simply adding another dozen references that agree with the author does not help improve an article which is starting to become overly long with too many references.
Author Response
Thanks for your feedback. I have read, and am now referring to, the article on rational egoism by Edwin Locke, and Jaana Woiceshyn. I have also elaborated a bit why I am not convinced that such a way of reasoning will prevent dishonesty in every situation.
Round 3
Reviewer 2 Report
I see that you have your own personal strongly held opinion regardless of what other scholars have said about ethical egoism or rational egoism. You have not altered your original claims in any substantive way based on my feedback or attempts to point out important contradictory works. You basically repeat your original claims regardless of any contradictory scholarly references.
I'm sure there are many suitable outlets for your opinions, but I do not think an academic or scholarly PRJ should be one of them.
Author Response
Thank you for this response. It's been helpful to chech out a number of references to ethical/rational egoism. I have tried to argue why egoism is a problem for Robust Moral Realism, but have not written that this problem is impossible to solve or that every version of ethical/rational egoism entails an acceptance of cruel or horrendous action. Further, I have also presented references that supports directly or indirectly my argument.